# A Rapid and Feasible ^1^H-NMR Quantification Method of Ephedrine Alkaloids in *Ephedra* Herbal Preparations

**DOI:** 10.3390/molecules26061599

**Published:** 2021-03-13

**Authors:** Hsin-Yi Hung, Shih-Min Lin, Chia-Ying Li, Sio-Hong Lam, Yu-Yi Chan, Meei-Jen Liou, Po-Chuen Shieh, Fu-An Chen, Ping-Chung Kuo, Tian-Shung Wu

**Affiliations:** 1School of Pharmacy, College of Medicine, National Cheng Kung University, Tainan 701, Taiwan; z10308005@email.ncku.edu.tw (H.-Y.H.); shlam@mail.ncku.edu.tw (S.-H.L.); 2Department of Chemistry, National Cheng Kung University, Tainan 701, Taiwan; lsming0716@gmail.com; 3Department of Applied Chemistry, National Pingtung University, Pingtung 900, Taiwan; cyli@mail.nptu.edu.tw; 4Department of Biotechnology, Southern Taiwan University of Science and Technology, Tainan 710, Taiwan; yuyichan@stust.edu.tw; 5Department of Applied Chemistry, Providence University, Taichung 433, Taiwan; mjliou@pu.edu.tw; 6Department of Pharmacy, College of Pharmacy and Health Care, Tajen University, Pingtung 907, Taiwan; pochuen@tajen.edu.tw (P.-C.S.); fachen@tajen.edu.tw (F.-A.C.)

**Keywords:** quantitative analysis, ephedrine, pseudoephedrine, *Ephedra*, ^1^H-NMR, internal standard

## Abstract

A highly specific and sensitive proton nuclear magnetic resonance (^1^H-NMR) method has been developed for the quantification of ephedrine alkaloid derivatives in *Ephedra* herbal commercial prescriptions. At the region of δ 4.0 to 5.0 ppm in the ^1^H NMR spectrum, the characteristic signals are separated well from each other, and six analogues in total, methylephedrine (ME), ephedrine (EP), norephedrine (NE), norpseudoephedrine (NP), pseudoephedrine (PE), and methylpseudoephedrine (MP) could be identified. The quantities of these compounds are calculated by the relative ratio of the integral values of the target peak for each compound to the known concentrations of the internal standard anthracene. The present method allows for a rapid and simple quantification of ephedrine alkaloid derivatives in *Ephedra*-related commercial prescriptions without any preliminary purification steps and standard compounds, and accordingly it can be a powerful tool to verify different *Ephedra* species. In comparison to conventional chromatographic methods, the advantages of this method include the fact that no standard compounds are required, the quantification can be directly performed on the crude extracts, a better selectivity for various ephedrine alkaloid derivatives, and the fact that a very significant time-gain may be achieved.

## 1. Introduction

The aerial parts of *Ephedra* have long been used as diaphoretics, antiasthmatics and diuretics, as well as for the treatment of bronchitis and acute nephritic edema, and to induce perspiration, reduce fever, and treat cough and asthma in traditional Chinese medicine. Ephedra-containing dietary supplements have been promoted for use as aids in dieting and as stimulants for boosting energy and athletic performance [1,2,3,4]. The activity of *Ephedra* species is attributed to the presence of optically active diastereomeric alkaloids (Figure 1), of which ephedrine (EP) and pseudoephedrine (PE) constitute the major fractions [4]. *Ephedra* plant materials used in oriental medicines show quite variable qualities since there are many species comprising the sources of the *Ephedra* on the market [5,6]. Moreover, the diverse geographical origins of the plant materials make the total content of the main active alkaloids quite different from species to species.

The International Olympic Committee listed ephedrine and related compounds as stimulants in athletic sports in 2003 [7]. However, considering that these products are often used for the treatment of colds, the International Athletic Committee has made a quantitative limitation for ephedrine alkaloids. In December 2003, the FDA issued a rule to declare dietary supplements that contained ephedrine alkaloids to be adulterated [8]. This rule was based on the evidences of health risks associated with the uses of *Ephedra* and in effect banned the use of ephedrine alkaloids (regardless of their botanical origin) in dietary supplements. In addition, *Ephedra* was used extensively in various traditional Chinese medicine prescriptions, including Ding Chuan Tang, Shern Mih Tang, Ma Huang Tang, and Ma Xing Yin Gan Tang, which aroused our interests due to the quality of these prescriptions. Increasing concerns about the safety of *Ephedra* alkaloids from both the consumers and regulatory agencies have led to the development of several methods to detect these analytes in a range of complex matrices.

A number of methods for the quantitative analysis of *Ephedra* alkaloids have been reported, including thin-layer chromatography (TLC) [9], gas-chromatography (GC) [10,11], GC-mass spectrometry (GC-MS) [12,13], high-performance liquid chromatography (HPLC) [14,15,16,17,18], and capillary electrophoresis (CE) [19]. HPLC had been extensively used for ephedrine alkaloids analysis, and in many cases SDS was applied into the mobile phase to improve the resolution. However, it also increased the difficulties in separating the amphiphilic compounds. The lack of a specific and strong chromophore for detection was also a problem when a conventional HPLC–UV detector was used. The GC method was usually admired as the most popular technique for the quantitation of ephedrine analogs. In order to enhance the sensitivity and to remove interference compounds, complex clean-up procedures and precolumn derivatization were required before analysis, leading to time-consuming protocols. Therefore, developing a simple and accurate method for the simultaneous detection of *Ephedra* alkaloids for the quality control of *Ephedra* raw materials and for commercial pharmaceutical prescriptions is strongly required.

Recently, ^1^H-NMR spectroscopy was developed as an important tool for the quality control of phytochemical preparations [20,21,22,23,24,25,26,27,28], clinical diagnosis, and monitoring of treatment [29]. The advantages of the quantitative NMR (qNMR) method are manifold. For example, it is rapid, noninvasive, and does not require any sample pretreatment steps. In addition, no standard compounds are required to prepare the calibration curves, and it detects all the components presented in herbal preparations simultaneously in a single measurement. Although there are still some defects for qNMR, such as a high cost of instruments and lower sensitivity compared to the traditional chromatographic methods, more and more reports related to the qNMR methods are published due to the lack of many certified reference materials. Therefore, in this study we described a ^1^H NMR spectroscopic method for the quantitative analysis of ephedrine alkaloid derivatives in *Ephedra* species and related commercial traditional Chinese medicine prescriptions, including Ding Chuan Tang, Shern Mih Tang, Ma Huang Tang, Ma Xing Yin Gan Tang, Ye Jiao Teng, and Gui Pi Tang. This method would allow for the rapid and simultaneous determination of these ephedrine alkaloid derivatives to be performed without any pretreatment steps.

## 2. Results and Discussion

The extraction of alkaloids was performed according to the reported method [30]. Considering the solubility of ephedrine alkaloids, CDCl_3_ was used as the solvent to ensure that all the extract could be dissolved. The ^1^H NMR spectra of the extracts of *E. sinica*, *E. intermedia*, and *E. equisetina* were well documented in CDCl_3_ (Figure 2), and the NMR signals of these alkaloids were provided in Table 1. For the quantitative analysis of these alkaloids, including methylephedrine (ME), ephedrine (EP), norephedrine (NE), norpseudoephedrine (NP), pseudoephedrine (PE), and methylpseudoephedrine (MP), the H-1 signals of these compounds were selected as the target signals since they were well separated in the region of δ 4.0–5.0 ppm and no significant interferences were observed (Figure 3). According to the literature reports and compared with the provided spectra, the target protons (H-1) of ephedrine and pseudoephedrine are resonated at δ 4.76 (d, *J* = 3.9 Hz) and 4.17 (d, *J* = 8.2 Hz) [31]. The proton signals of methylephedrine and methylpseudoephedrine are also well separated from each other and observed at δ 4.96 (d, *J* = 3.8 Hz) and 4.19 (d, *J* = 6.4 Hz) [32]. For the third pair of diastereomers, norephedrine and norpseudoephedrine, the H-1 signals are located at δ 4.52 (d, *J* = 4.8 Hz) and 4.24 (d, *J* = 6.8 Hz) [31,33], respectively. Therefore, these data suggest that the H-1 signal is suitable for use as a target peak for quantification.

A suitable internal standard should preferably be a stable compound with a signal in a noncrowded region of the ^1^H-NMR spectrum. For this purpose, anthracene, with a signal at δ 8.41 and the integral value staying constant within 48 h, has been designated. In the case of the qNMR analysis, calibration curves of standard compounds were not necessary for quantification since the integration ratios between any proton signals and internal standard were always proportional to their concentration ratio. Therefore, the unknown concentrations of ephedrine alkaloids in the tested samples could be afforded by the simple calculation of the integral area ratio between anthracene and each target peak. The NMR signals can be improved by increasing the number of scans, and the validation parameters may therefore be varied. However, the ephedrine standard was applied to check the accuracy and limit of detection in the current NMR parameters. The accuracy of this method was determined by adding a known concentration of ephedrine (0.5, 1.0, 2.0 mg, respectively) to extract samples. The peak area corresponding to ephedrine was found to increase proportionally to the added concentration of the standard, and the average recovery percentage was 96.5% (three samples measured in triplicate). The limit of quantification (LOQ) and limit of detection (LOD) for ephedrine under the present experimental condition was determined to be 0.2 and 0.05 mg/mL at the signal-to-noise ratios of 10 and 3, respectively. Since we did not have the standards for other alkaloids, the LOD and LOQ of these compounds were not examined.

Moreover, an HPLC-UV analysis of the ephedrine standard was also performed. The calibration curve for ephedrine was established in the range between 0.2 and 2.0 mg/mL and possessed good linearity (R^2^ = 0.9991) in order to compare the results afforded by the NMR method with those from the chromatographic technique. The examined samples exhibited comparable data with these two analytical methods (data not shown). Compared with the previously reported data [16], the ephedrine content in *E. sinica* was in the range of 5–10 mg/g and was also comparable with the present analytical results.

Three *Ephedra* materials, including *E. sinica*, *E. intermedia,* and *E. equisetina,* were analyzed for the contents of ephedrine alkaloids using this ^1^H-NMR method. In the ^1^H-NMR spectra of these extracts, the H-1 signals at δ 4.96, 4.76, 4.52, 4.24, and 4.17 ppm for ME, EP, NE, NP, and PE, respectively, were well separated from other signals. These H-1 peaks were well assigned, according to the literature reports [31,32,33]. However, the lack of H-1 signal of MP indicated that the quantity of MP in *Ephedra* materials was low. These results were identical to the reports [15,16,17,18] and were further confirmed by using the published method [16], which showed that the amounts of alkaloids in the *E. sinica* extract were in the order of EP > PE > ME > NP > NE, while MP was trace. The quantification of these five alkaloids by ^1^H-NMR was feasible by the calculation of the ratio of the integral area between well-separated specific proton (H-1) signals of these compounds and the internal standard. The quantitatively analytical data of ME, EP, NE, NP, and PE in three *Ephedra* species determined by this ^1^H-NMR method are shown in Table 2. Ephedrine (EP) was found to be the major constituent in *E. sinica*, and the amount of EP was about two to three times that of pseudoephedrine (PE). Meanwhile, in *E. intermedia*, the major constituent was PE, and its amount was about two to three times that of EP. In addition, EP was the major component in *E. equisetina,* and other alkaloids were relatively few.

This developed method can be applied for the identification of the presence of *Ephedra* species in related commercial traditional Chinese medicinal preparations (Table 2). In the present study, all the *Ephedra* prescriptions (Ding Chuan Tang, Shern Mih Tang, Ma Huang Tang, and Ma Xing Yin Gan Tang purchased from companies A–D) were identified as containing *E. sinica* as their source rather than *E. intermedia* or *E. equisetina,* since ephedrine and pseudoephedrine were the major principles in their ^1^H NMR spectra (Figure 4a–d), which were identical with the profile of *E. sinica*. The commercial *Ephedra* prescriptions Ding Chuan Tang produced by different companies (A~D) showed similar contents of the ephedrine alkaloid derivatives. The prescriptions Shern Mih Tang showed a significantly different composition of ephedrine alkaloid derivatives. In particular, in the prescriptions provided by company C, the contents of ephedrine alkaloid derivatives were about two times higher than those in the prescriptions produced by companies A, B, and D. Similar situations were observed in the Ma Huang Tang prescriptions. The contents of ephedrine alkaloid derivatives in the prescription produced by company D were two times higher than those provided in companies A, B, and C. In addition, the alkaloid contents in Ma Xing Yin Gan Tang produced by companies A and D were somewhat lower than those in the other two companies (B and C). These variations may be due to the use of different stock materials by different companies or the abnormal addition of *Ephedra* plant materials to enhance the activity. Therefore, the developed ^1^H-NMR method met the analytical criteria for commercial *Ephedra* prescriptions, and the results were reproducible for *Ephedra* pharmaceutical preparations from various sources.

With the assistance of this ^1^H-NMR method, it would be efficient to check the adulterates of the ephedrine alkaloid derivatives or *Ephedra* materials in traditional Chinese medicine prescriptions. For example, the ephedrine alkaloid derivatives should not be detected in the prescriptions Ye Jiao Teng and Gui Pi Tang since there were not any *Ephedra* plant materials included. Accordingly, the prescriptions from companies A, B, and C did not show any signals at the region of δ 4.0 to 5.0 ppm in their ^1^H-NMR spectra, indicating that these prescriptions do not contain ephedrine alkaloid derivatives. However, the prescription Ye Jiao Teng produced by company E showed significant appearances of ephedrine (EP) and pseudoephedrine (PE) (1.3091 mg/g for EP and 1.4822 mg/g for PE), indicating that it may be adulterated by synthetic products or raw materials (Figure 5). Similarly, the ^1^H-NMR spectra of the prescription Gui Pi Tang produced by company E also displayed some quantities of EP (0.2148 mg/g) and PE (0.1236 mg/g) at a ratio of about 2:1 (Figure 5). This composition was similar to the profile of *E. sinica* and suggested that Gui Pi Tang of company E was adulterated with the plant materials of *E. sinica* or that the prescription may have been cross-polluted by the *Ephedra* preparations in the production processes.

## 3. Materials and Methods

### 3.1. Chemicals and Materials

First grade sulfuric acid (98%), ether, methanol, sodium chloride, and anthracene were purchased from E. Merck (Darmstadt, Germany). CDCl_3_ (99.9%) was obtained from Aldrich (Milwaukee, WI, USA). The reference compound ephedrine was isolated from the stems of *E. sinica* in our previous study, and its purity was checked by the NMR and HPLC methods (>99.5%). The plant materials of *E. sinica*, *E. intermedia,* and *E. equisetina* were purchased at an herb shop in Tainan (August 2005) and authenticated by Prof. C. S. Kuoh (Department of Life Sciences, National Cheng Kung University, Tainan, Taiwan). The commercial traditional Chinese medicine prescriptions (Ding Chuan Tang, Shern Mih Tang, Ma Huang Tang, and Ma Xing Yin Gan Tang) were purchased from four pharmaceutical companies (A–D) in Taiwan. In addition, Ye Jiao Teng and Gui Pi Tang were also purchased from four pharmaceutical companies (A–C and E) in Taiwan.

### 3.2. Sample Extraction

The extraction of alkaloids was performed according to the method of Cui et al. [30] with minor modifications. 0.4 g of powdered plant material or 1.0 g of prescription was weighed and transferred to a 15-mL centrifuge tube and mixed with 8 mL of 0.5 M H_2_SO_4_ solution. The mixture was shaken and sonicated for 1 h and then centrifuged for 20 min at 3000 rpm. The supernatant layer was transferred at 4.0 mL to another tube and 1.2 mL of 5 M KOH solution, 2.4 g NaCl and 5 mL of diethyl ether were added to the tube. The mixture was shaken for 20 min and then centrifuged for 20 min at 3000 rpm. Extraction with diethyl ether was performed twice. Diethyl ether layers were combined and evaporated after the addition of internal standard. The dried sample was dissolved in 1 mL of CDCl_3_ and used for the ^1^H-NMR measurement.

### 3.3. NMR Analysis and Identification of Ephedrine Alkaloid Derivatives

^1^H-NMR spectra were recorded in CDCl_3_ (99.9%) by using a Varian UNITY plus 400 MHz NMR spectrometer equipped with a Varian Indirect NMR™ (ID) probe. Each sample was dissolved by deuterated solvent in a Schotte economic NMR sample tube (5 mm o.d./178 mm length). For each sample, 128 scans were recorded with the following parameters: 0.187 Hz/point; spectra width, 3600 Hz; a 90° pulse was used to obtain the maximum sensitivity; relaxation delay, 10 s; acquisition time, 2.56 s. For the quantitative analysis, the peak area was used, and the start and end points of the integration of each peak were selected manually.

### 3.4. Recovery, Limit of Quantification (LOQ), and Limit of Detection (LOD) of Ephedrine

Recovery tests were selected to determine the accuracy of the method, in which three different concentrations of ephedrine (0.5, 1.0, 2.0 mg, respectively) were spiked into the sample and the recovery percentages were calculated using the measured contents divided by the contents of added standards and original sample obtained by ^1^H-NMR analysis. A blank recovery sample was prepared and analyzed for the comparison. The LOQ and LOD for ephedrine under the present NMR analytical condition (128 scans) were determined at signal-to-noise ratios of 10 and 3, respectively.

## 4. Conclusions

The present ^1^H-NMR method allows for the rapid and simultaneous determination of six alkaloids in *Ephedra* species, including methylephedrine (ME), ephedrine (EP), norephedrine (NE), norpseudoephedrine (NP), pseudoephedrine (PE), and methylpseudoephedrine (MP), without any pretreatment steps. With the assistance of this technique, the contents of the ephedrine alkaloid derivatives can be analyzed within a much shorter time than with conventional chromatographic methods. These alkaloids can be detected without any preliminary derivatization steps. The overall profiles of the plants or prescriptions can be quickly obtained, and it would therefore be a powerful tool for verifying different species. Hopefully, this method can be applied in the near future to the quality control of commercial pharmaceutics or preparations of *Ephedra* such as cough syrups and dietary supplements.

## Figures and Tables

**Figure 1 molecules-26-01599-f001:**
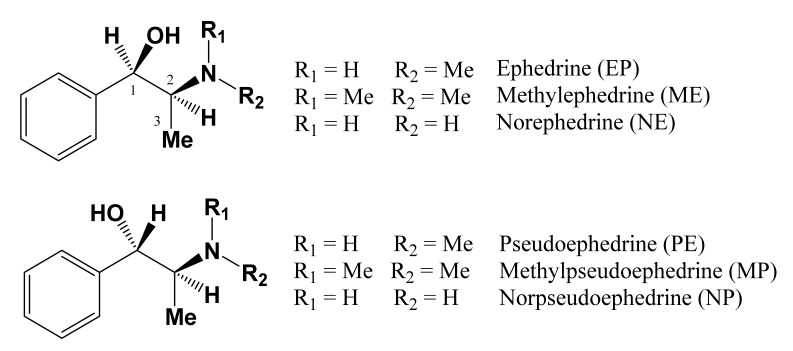
Structures of the active alkaloids in *Ephedra* species.

**Figure 2 molecules-26-01599-f002:**
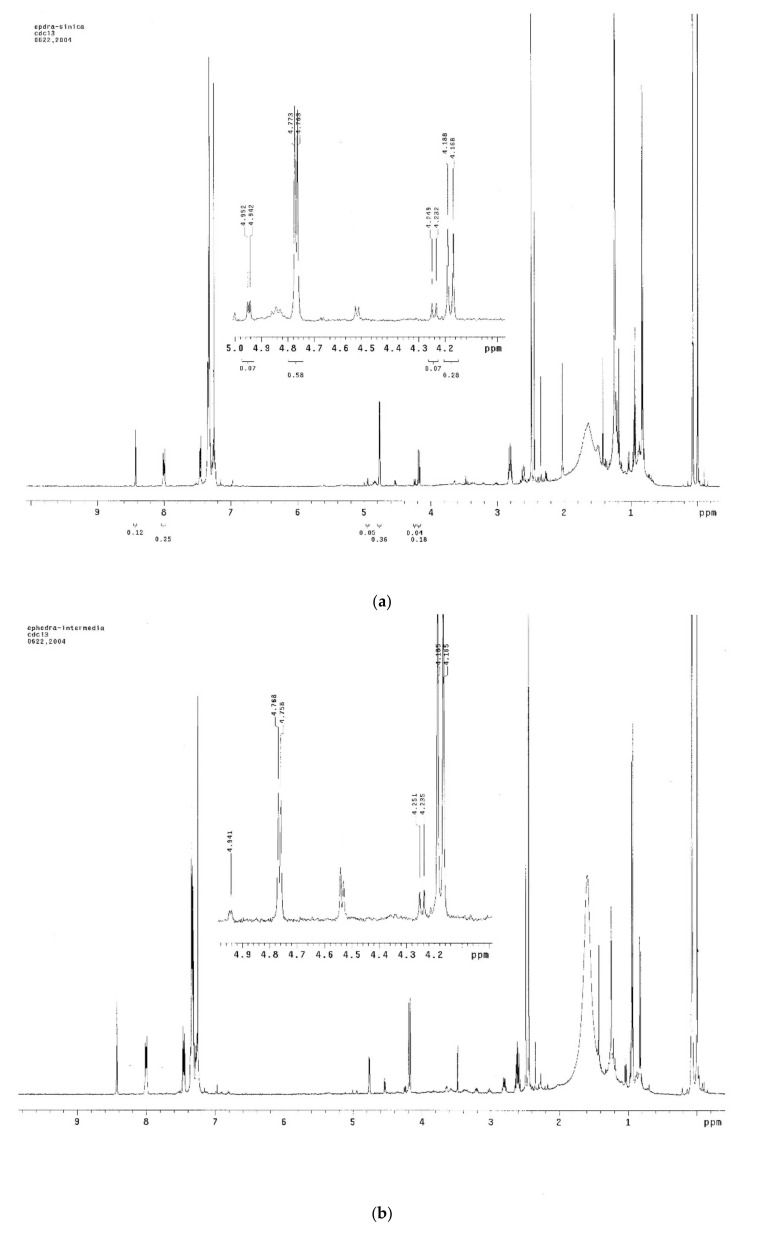
The ^1^H-NMR spectra of three *Ephedra* extracts. (**a**) *E. sinica* (ES); (**b**) *E. intermedia* (EI); and (**c**) *E. equisetina* (EE).

**Figure 3 molecules-26-01599-f003:**
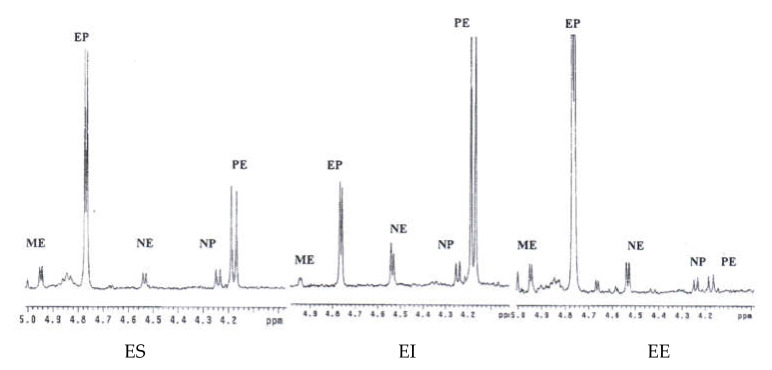
The H-1 signals of ephedrine alkaloids in the ^1^H-NMR spectra of three *Ephedra* extracts [*E. sinica* (ES), *E. intermedia* (EI), and *E. equisetina* (EE)].

**Figure 4 molecules-26-01599-f004:**
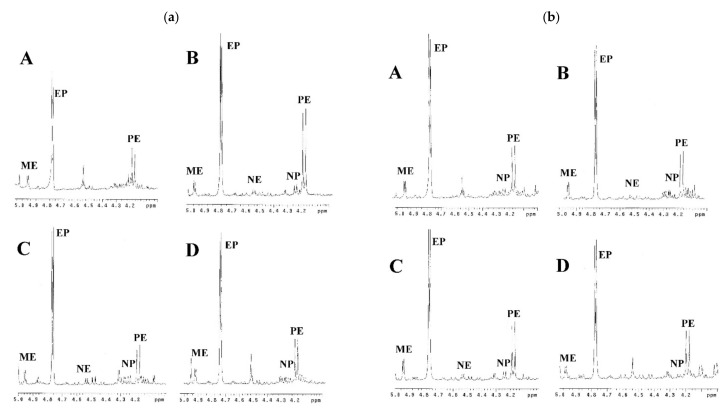
^1^H-NMR spectra of the *Ephedra* commercial prescriptions in the range of δ 4.0~5.0 ppm. (**a**) Ding Chuan Tang, (**b**) Shern Mih Tang, (**c**) Ma Huang Tang, and (**d**) Ma Xing Yin Gan Tang. A~D: Samples from companies A~D.

**Figure 5 molecules-26-01599-f005:**
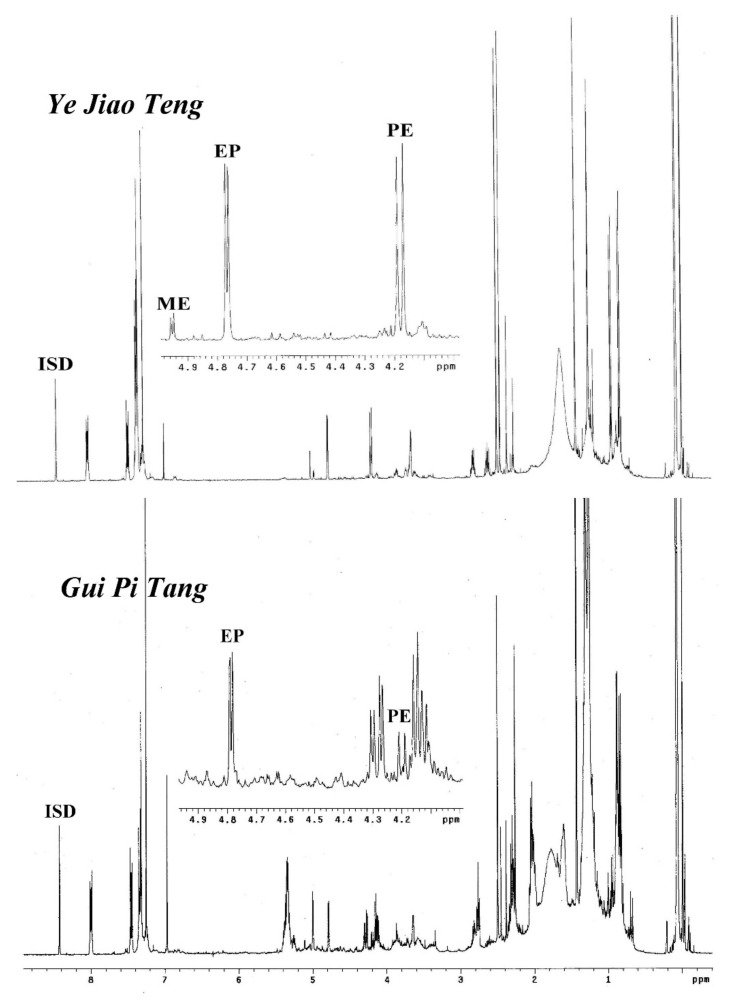
^1^H-NMR spectra of the commercial prescriptions Ye Jiao Teng and Gui Pi Tang from company E. ISD: anthracene.

**Table 1 molecules-26-01599-t001:** ^1^H-NMR signal assignments for the ephedrine alkaloid derivatives (δ in ppm).

Compound	Aromatic Protons	H-1	H-2	CH_3_-3	NCH_3_
methylephedrine (ME)	7.14–7.31	4.96	2.45	0.74	2.28
ephedrine (EP)	7.32	4.76	2.65	0.84	2.45
norephedrine (NE)	7.31	4.52	3.18	0.96	-
norpseudoephedrine (NP)	7.21–7.38	4.24	3.03	1.04	-
pseudoephedrine (PE)	7.33	4.17	2.63	0.95	2.45
methylpseudoephedrine (MP)	7.20–7.43	4.19	2.59	0.72	2.31

**Table 2 molecules-26-01599-t002:** The concentrations (mg/g) of ephedrine alkaloid derivatives, including methylephedrine (ME), ephedrine (EP), norephedrine (NE), norpseudoephedrine (NP), pseudoephedrine (PE), and methylpseudoephedrine (MP), in *E. sinica*, *E. intermedia*, and *E. equisetina* extracts and several *Ephedra*-related commercial prescriptions (companies A~E) *^a^*.

Sample	Source	ME	EP	NE	NP	PE
*E. sinica*	A	0.9931(0.0291) *^b^*	8.4931(0.0881)	0.2819(0.0007)	0.9963(0.0021)	4.0951(0.0113)
*E. interdemia*	A	0.3812(0.0003)	2.0461(0.0115)	0.3402(0.0010)	0.9332(0.0023)	4.9088(0.0098)
*E. equisetina*	A	0.7607(0.0012)	19.3561(0.0221)	0.3361(0.0009)	0.5183(0.0013)	0.6643(0.0021)
Ding Chung Tang	A	0.4384(0.0007)	1.9869(0.0038)	ND *^c^*	ND	0.7676(0.0006)
	B	0.2330(0.0012)	2.0349(0.0017)	0.0572(0.0009)	0.1768(0.0002)	1.1146(0.0013)
	C	0.2355(0.0004)	1.8970(0.0001)	0.0646(0.0003)	0.1471(0.0004)	0.6468(0.0016)
	D	0.2520(0.0014)	1.6781(0.0028)	ND	0.1407(0.0017)	0.5689(0.0005)
Shern Mih Tang	A	0.5868(0.0003)	3.9408(0.0004)	ND	0.3108(0.0007)	1.5428(0.0006)
	B	0.4391(0.0009)	3.5641(0.0003)	0.1446(0.0003)	0.1970(0.0002)	1.2378(0.0001)
	C	0.8775(0.0004)	6.5161(0.0027)	0.2309(0.0004)	0.4475(0.0004)	2.5308(0.0005)
	D	0.2966(0.0024)	2.4228(0.0007)	ND	0.1748(0.0005)	1.0037(0.0003)
Ma Huang Tang	A	1.1414(0.0006)	8.5055(0.0011)	0.3140(0.0001)	0.4607(0.0008)	3.3497(0.0010)
	B	0.5860(0.0002)	4.4837(0.0031)	ND	0.2644(0.0007)	1.3534(0.0008)
	C	0.6648(0.0002)	5.7699(0.0082)	0.2121(0.0002)	0.5076(0.0002)	2.2410(0.0002)
	D	0.7731(0.0006)	5.2176(0.0005)	0.2000(0.0004)	0.3935(0.0005)	1.5321(0.0015)
Ma Xing Yin Gan Tang	A	0.2661(0.0002)	2.0493(0.0047)	0.1875(0.0002)	0.2363(0.0005)	1.2844(0.0036)
	B	0.4271(0.0002)	4.0306(0.0007)	0.3563(0.0003)	0.2857(0.0002)	1.4700(0.0006)
	C	0.4968(0.0002)	4.3671(0.0004)	0.1450(0.0017)	0.3337(0.0006)	1.6396(0.0006)
	D	0.2562(0.0003)	1.2551(0.0007)	0.0785(0.0001)	0.1357(0.0003)	0.4783(0.0019)
Ye Jiao Teng	A	ND	ND	ND	ND	ND
	B	ND	ND	ND	ND	ND
	C	ND	ND	ND	ND	ND
	E	0.2322(0.0005)	1.3091(0.0010)	ND	ND	1.4822(0.0011)
Gui Pi Tang	A	ND	ND	ND	ND	ND
	B	ND	ND	ND	ND	ND
	C	ND	ND	ND	ND	ND
	E	ND	0.2148(0.0002)	ND	ND	0.1236(0.0005)

*^a^* Recorded on mg/g of material. *^b^* All the experiments were performed in triplicate (*n* = 3). Results were expressed as the mean (standard deviation). *^c^* Not detected.

## Data Availability

The data presented in this study are available on request from the corresponding author.

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
