# Peer review of "A Rapid and Feasible 1H-NMR Quantification Method of Ephedrine Alkaloids in Ephedra Herbal Preparations"

_molecules, 2021, doi:10.3390/molecules26061599_

Round 1
Reviewer 1 Report
This manuscript presents the quantification of ephedrin in plant extracts by NMR. This would certainly be a useful resource to be developed, however this study is certainly falls shat of this goal and cannot be published as it is.
The idea of using the 4-5 ppm region to recognize, separate and quantify the different compounds of the ephedrine family is valid, however validating the approach by measuring a couple of good and bad samples is simply insufficient.
To the least, it should be checked which kind of compound might appear in this zone, either by exploring a few type of plant and food extracts, or by checking in chemical shift databases.
Without this information, the evaluation may be seriously biased and compromised.
Also , the NMR approach could be compared to a standard ephedrine quantification method.
In addition, this work presents many defects that prevent its publication.
The NMR acquisition conditions are by no mean correct for quantitative measurements. A relaxation delay of 1 sec in CDCl3 is much too short. In addition a pulse of 4µsec is meaningless, the flip angle should be provided.
The concentration range used for calibration is 0.2-2 mg/ml but the LOD is said to be 0.05mg/ml - You do not do validation that way ! this is not possible.
The spectra are of extremely poor quality, and appears to be rapid scans of drawn NMR spectra. This is in no way acceptable. The text in the figures are readable (integrals, axes, etc.)
In addition, the limits of the spectral excerpts are not constant, making comparisons difficult.
P4 l96 It is well established that diastereoisomers are distinguishable by NMR, and should not be a subject of surprise, as the two chiral centres are in geminal positions.
p2 l50-55 the text mention matrices (Food, etc.) which are never tested further in the work.
fig 1 "Peudeephedrine"
Finally, the language is poor, to the point that some sentences are barely understandable. The text should be carefully corrected by an English native speaker.
Author Response
Review attached

Reviewer 2 Report
The authors present a 1H NMR method developed to quantify ephedrine alkaloid derivatives in the Ephedra herbal commercial prescriptions.
In my opinion, the manuscript presents points to be considered:
1.- The figures of the spectra provided by the authors have low quality, and the sharpness of the peaks it is not acceptable.
2.- The NP doublet is very close to PE one; thus, a minimum deformation of the signals can compromise the quantification.
3.- It is necessary to perform the validation of the proposed method. Besides recovery and LOD, limit of quantification, linear range, working range, and robustness must be reported.
4.- The advantages of the NMR method is claimed, but he quantification of the alkaloids also should be performed using another method (HPLC-MS), because it is necessary to contrast the results obtained with each method.
In conclusion, the manuscript could be suitable for publication in Molecules if the authors attend the recommendations.
Author Response
Review attached

Reviewer 3 Report
The submitted manuscript presents the results of the well designed qNMR study. It is a significant contribution in the field of pharmaceutical analysis. However, there are some major queries, listed below, that prevent me from accepting this paper. I recommend major revision.
Lines 76-78, if you mention the advantages of qNMR you should also discuss its disadvantages. Don’t get me wrong-I admire qNMR and use it a lot but to be honest those cons of qNMR should be at least mentioned (i.e. high cost of the analysis, lower sensitivity when compared to chromatographic methods etc.).
The quality of Figure 2 is very poot, you must improve it.
The quality of Figures 3-5 is also very poor, really, improve dpi of the spectra. Besides, in some spectra (i.e. Figure 4c) A and B) the x-axis labels are missing.
The Authors should create additional Table presenting full 1H NMR signal assignments for the studied compounds (for all of the H atoms). It would enable the Reader to confirm if the choice of 1-H was correct.
Line 121, what about LOQ?
Table 1, though it is mentioned earlier in the text, the abbreviations (ME, EP, …) must be also defined in the caption to this table.
Line 137, this sentence is grammatically incorrect. The whole manuscript should be checked by a native speaker.
Line 147, it should be “major component”
Table 1, for the sources of some preparations you use “A, B, C, D” and for the others “A, B, C, E”. Does is mean that, for example, A for Ding Chung Tang and A for Ma Huang Tang is the same manufacturer? It should be clarified.
Author Response
Review attached

Round 2
Reviewer 1 Report
The authors do not have substantially modified the manuscript with respect to my remarks from the first version.
I still recommend rejection
several of the main points I raised in the first report are still very problematic
- my point 2 in the first version, about the fact the authors mention a relaxation delay of 1sec and do not mention the flip angle but a pulse duration of 4µs. Which is both insufficient information, and points to very problematic acquisition conditions incompatible with qNMR.
Now - without changing the spectra figures (so not having redone the experiments), they mention a relaxation delay of 10 sec ? How can that be ? To the least, it manifests a very lazy approach to the publication, and perhaps a more problematic honesty problem.
- my point 3, about a LOD of 0.05mg/ml while the lower concentration for the experimental scheme is 0.2 mg/ml - this has not been modified
This is not a grammar error! as the authors seem to believe in their answer - it is an error in the conclusion.
- my point 4 about the very low quality of the figures both in design (varying spectral windows) and in rendering (much below the minimum quality required).
Nothing has changed.
- my point 1 about the 4-5 ppm region - a simple bibliography study is not really enough. At the very least a check in chemical shift databases could point to potential errors.
- but Ok I can live with this one -
Reviewer 2 Report
The authors improved the discussion and the experimental sections; however, it is necessary they include in the manuscript:
1.- The characteristics of the used probe and the quality of the NMR tubes.
2.- A brief discussion about the contrast between the data obtained from the previous HPLC analysis of the samples and the data obtained by NMR quantification.
In conclusion, the manuscript could be suitable for publication in Molecules if the authors attend the recommendations.
Reviewer 3 Report
Though the language should be polished even more, I guess the technical editors will help you. From my point of view this manuscript is suitable for publication.
Author Response
Thank you for your comment. We wish to express our sincere thanks for your appreciation.
Round 3
Reviewer 1 Report
This version is improved
About Point 2, (relaxation delay) the authors explanations can be received, and I want to believe in their good faith.
Point 3 (LOD) they now better explain their procedure, which is a common one in qNMR. This could be discussed, but this paper is not the place. So ok.
point 4 (figure quality) the quality is still below what it should be (jpg scans with a lot of visible artefacts) but at least you can read them - so ok
So I believe it is ok to publish this !
Reviewer 2 Report
The authors have improved the manuscript, but the information about the used probe still remains missing (ID?); however I consider they can add this informaton easly. Therefore, I consider the manuscript, with this little improvement, it becomes suitable for publication in Molecules.